# Intracellular Polyphenol Wine Metabolites Oppose Oxidative Stress and Upregulate Nrf2/ARE Pathway

**DOI:** 10.3390/antiox11102055

**Published:** 2022-10-19

**Authors:** Chiara Stranieri, Flavia Guzzo, Sofia Gambini, Luciano Cominacini, Anna Maria Fratta Pasini

**Affiliations:** 1Department of Medicine, Section of Internal Medicine D, University of Verona, 37134 Verona, Italy; 2Department of Biotechnology, University of Verona, 37134 Verona, Italy

**Keywords:** metabolomics, Nrf2/ARE, oxidative stress, wine polyphenols

## Abstract

Moderate wine consumption has been associated with several benefits to human health due to its high polyphenol content. In this study, we investigated whether polyphenols contained in a particular red wine, rich in polyphenols, can pass the cell membrane and switch the oxidant/antioxidant balance toward an antioxidant pattern of THP-1 cells and human cardiomyocytes through a gene regulatory system. First, we identified which metabolite polyphenols present in red wine extract cross cell membranes and may be responsible for antioxidant effects. The results showed that the wine metabolites in treated cells belonged mainly to stilbenes, flavan-3-ols derivatives, and flavonoids. Other metabolites present in cells were not typical wine metabolites. Then, we found that red wine extract dose-dependently lowered reactive oxygen species (ROS) induced by tert-butyl hydroperoxide (TBHP) up to 50 ± 7% in both cell lines (*p* < 0.01). Furthermore, wine extract increased nuclear Nrf2 of about 35 ± 5% in both cell lines (*p* < 0.01) and counteracted its reduction induced by TBHP (*p* < 0.01). The rise in Nrf2 was paralleled by the increase in hemeoxygenase-1 and glutamate-cysteine ligase catalytic subunit gene expression (both mRNA and protein) (*p* < 0.01). These results could help explain the healthful activity of wine polyphenols within cells.

## 1. Introduction

Moderate wine consumption has been associated with several benefits to human health, including the prevention of cardiovascular diseases [1], diabetes, cancer, and neurological disorders [1,2,3]. Interestingly, moderate wine consumption has been included among the healthy lifestyle factors for the primary prevention of coronary heart disease [4] and heart failure [5]. Wine, an essential component of the Mediterranean diet [6], is different from other alcoholic beverages because many of its described health-promoting activities are independent of its alcohol content and are positively correlated with its polyphenolic content [7,8,9,10,11]. Remarkably, it has been suggested that moderate red wine drinkers may consume well above average polyphenol levels [12]. The red wine phenolics are commonly divided into two major groups: flavonoids and non-flavonoids. The main flavonoid compounds present in red wine comprise several classes, such as flavanols, flavonols, and anthocyanins, whereas non-flavonoid compounds present in wine include phenolic acids, phenols, and stilbenes [11]. Polyphenol composition and exact content in wines are dependent on several factors; recent advances in viticulture allow oenologists to seek grapes with special characteristics, which not only include an adequate amount of sugars and acids, but also appropriate levels of other important grape components, such as phenolic compounds and terpenoids [13]. It is well known that the way wine flavonoids are absorbed and metabolized may influence their bioavailability, and consequently, their health-promoting effects [11]. There is now an available body of information suggesting that the molecules responsible for these effects are probably not the ingested ones, but are metabolites that arise after polyphenol absorption or after the activity of microbiota [14]. In fact, although the majority of polyphenols are carried into the large intestine where bacteria catabolize them to chain fission products and/or monomeric catechins [13,14], some of the monomeric polyphenols and oligomeric tannins are absorbed in humans, with or without deglucosylation, and enter into the blood circulation by going across the epithelial cells in the small intestine [13,14,15,16]. Moreover, several studies indicate that monomers and dimeric procyanidins can be carried inside cells and directly modulate the activity of signaling proteins and/or prevent oxidation [17]. On the contrary, bigger and nonabsorbable procyanidins may regulate cell signaling by interacting with cell membrane proteins and lipids, producing changes in membrane biophysics, and modulating oxidant production [17]. 

Oxidative stress is considered an imbalance between the generation of reactive oxygen species (ROS) and their removal by defensive mechanisms [18]. Oxidative stress can trigger a series of transcription factors, which contribute to the expression of some genes associated with inflammation [19]. In particular, oxidative stress has been shown to activate IĸB kinase that causes the phosphorylation of IĸB-α, resulting in nuclear translocation of the nuclear factor (NF)-κB with subsequent transcription of pro-inflammatory cytokines [10]. The inflammation triggered by oxidative stress is the starting point of many chronic diseases [20]. 

Polyphenols have been suggested to be beneficial as adjuvant therapy for their antioxidant [21] and anti-inflammatory activities [22], as well as their inhibition of enzymes connected with the production of eicosanoids [23]. Polyphenols may, in addition, activate the transcription factor nuclear factor-E2-related factor 2 (Nrf2) [24,25], which controls the basal and induced expressions of an array of antioxidant response element (ARE)-dependent genes, such as heme-oxygenase (HO)-1 and glutamate-cysteine ligase catalytic (GCLC) subunit, to regulate the physiological and pathophysiological outcomes of oxidant exposure [26,27]. Under basal conditions, Nrf2-dependent transcription is repressed by its negative regulator Kelch-like enoyl-CoA hydratase-associated protein 1 (Keap-1); when cells are exposed to oxidative stress or electrophiles, Nrf2 accumulates in the nucleus and drives the expression of its target genes [26]. Recent reports have demonstrated that knocking out the Nrf2 gene makes mice highly susceptible to oxidative stress and inflammation in a wide variety of disease models affecting the liver, lungs, and cardiovascular systems [28,29,30,31,32]. Recent studies indicate that there is crosstalk between the Nrf2 and NF-ĸB pathways and evidence of Nrf2 negatively regulating the NF-kB signaling pathway [25]. 

Although several studies have shown the in vitro antioxidant activity of some red wine flavonoids [19,33], it has been recently suggested that most of the studies carried out using the native unmodified forms of flavonoids present in red wine should be interpreted with caution [11]. Moreover, while single phytochemicals, mostly tested at supraphysiological concentrations, were studied in cell culture and animal models for their beneficial effects to switch cell signaling pathway toward an antioxidant signature [34], no studies so far have evaluated the effects of polyphenols contained in wine extracts on cell oxidative stress and antioxidant signaling pathways. In this study, we investigated whether polyphenols contained in a particular red wine made from grapes and winemaking techniques designed to achieve a high content of polyphenols could pass the cell membrane and switch the oxidant/antioxidant balance toward an antioxidant pattern in THP-1 cells and human cardiomyocytes through a gene regulatory system. First, using ultraperformance liquid chromatography-electrospray ionization-mass spectrometry (UPLC-MS) analysis, we were able to identify which metabolite polyphenols present in red wine cross cell membranes and may be responsible for antioxidant effects within cells. Then, we discovered that red wine extracts reduce oxidative stress in cells and modify the expression of Nrf2/ARE genes linked to oxidative-stress-related signaling pathways, either in basal or oxidative stress-induced conditions.

## 2. Materials and Methods

### 2.1. Concentration of Polyphenolic Compounds and Wine Extract Preparation

For this study, we used a particular Italian red wine (Vitis Vitae, Revino Italia, Corte Ferrazzette, S. Martino B.A., Verona, Italy) made from grapes with winemaking techniques designed to achieve a high content of polyphenols. Several polyphenolic families were measured using a colorimetric method: total polyphenols, catechins, anthocyanins, and proanthocyanins. Total polyphenol content was estimated using the Folin–Ciocalteu method [35], which determines -OH groups based on the fact that light absorption increases as the number of -OH groups in a sample increases. Phenolic compounds react with Folin–Ciocalteu’s reagent only under basic conditions (adjusted by a sodium carbonate solution to pH = 10). Total phenolic content was calculated as a Gallic acid equivalent based on the standard curve with a Gallic acid standard prepared in corresponding conditions [35]. Total flavonoids (+catechin), non-anthocyanin flavonoids (+catechin), (+) catechin, and (−) epicatechin were spectrophotometrically measured on a SP65 ultraviolet (UV)/Visible spectrophotometer (Gallenkamp, Cambridge, UK), using a 1.0 cm optical path length glass [36].

The wine used for cell culture experiments was concentrated through Sep-Pak^®^ Plus C18 Cartridges (Waters, Milford, MA, USA). The cartridges were first conditioned with ethanol for 15 min and then equilibrated with water (2 × 10 mL). Then, 5 mL of wine were loaded onto the cartridges and washed with water (2 × 10 mL). Finally, the wine was eluted in 1 mL of methanol. The methanol was evaporated using a SpeedVac vacuum concentrator. The dried pellets were reconstituted with 1 mL hydroalcoholic solution (1:10) before use. 

### 2.2. Cell Cultures 

The macrophage-like THP-1 cell line was cultured, as previously described [37]. Human cardiomyocyte ventricular primary cells (Celprogen, Torrance, CA, USA) were cultured following the recommended manufacturer’s protocols in extracellular matrix pre-coated flasks and/or well plates with a ready-to-use complete growth medium (Celprogen, Torrance, CA, USA), as previously described [37]. The choice of such cell lines was guided by previous evidence of their high reliability in evaluating oxidative stress and antioxidant signaling pathways [38,39]_._

### 2.3. Bioavailability of Wine Extract Metabolites in THP-1 Cells

To assess which wine metabolites were bioavailable in cultured cells, we performed preliminary experiments by incubating different amounts of THP-1 cells (5, 10, and 20 million) with wine extract at a fixed concentration of 1 mg/mL of total polyphenols overnight (ON), and extensively washed them before metabolite extraction. After ON incubation, cells were washed 3 times with NaCl. The last wash was performed in 2 mL Eppendorf, the surnatant was aspirated, and the pellet stored at −80 °C in liquid nitrogen. For the experiments, THP-1 cells were slowly thawed in ice; thereafter, 700 µL of cold methanol and 700 µL of cold chloroform were added to the cells. The samples were vortexed for 30 s and sonicated for 10 min in an ice bath. Then, 466 µL of water was added, the samples were vortexed for 30 s, and subsequently centrifuged for 10 min at 21,000 g and 4 °C. The methanolic phases were recovered and analyzed.

### 2.4. UPLC-MS Analysis

For the UPLC-MS analysis, the methanol extracts were diluted 1:1 (*v*/*v*) with LC-MS grade water (Honeywell, Seezle, Germany), whereas a sample of concentrated wine, resuspended in 1 mL of LC-MS grade water, was diluted 1:10 (*v*/*v*). The diluted samples were passed through Minisart RC4 filters with 0.2 µm pores (Sartorius, Göttingen, Germany), and then 1, 3, and 5 µL of the cell extracts and 1 µL of the resuspended wine were injected into the UPLC-MS system. The analysis was performed with an ACQUITY I CLASS UPLC system (Waters Corporation, Milford, Massachusetts ), connected to a Xevo G2-XS qTOF mass spectrometer (Waters) equipped with an electrospray ionization (ESI) source operating in either positive or negative ionization modes. The chromatographic conditions and mass spectrometer parameters were set as described in Commisso et al. [40]. The raw data were processed with Progenesys QI (Nonlinear dynamics, UK). The metabolite identification was performed using an “in house” library of mass spectra, through the m/z value, and, where possible, isotopic similarity and fragmentation patterns. 

The data matrix obtained by using Progenesys QI was submitted to principal component analysis (PCA) through SIMCA 13.0 (Umetrics, Sartorius, Gottingen, Germany), after Pareto scaling and centering. 

### 2.5. Intracellular ROS Measurement

To induce different degrees of oxidative stress, THP-1 cells were incubated with increasing concentrations (20–100 μM) of tert-butyl hydroperoxide (TBHP) for 45 min at 37 °C. Intracellular ROS formation was determined using the CellROX Deep Red Flow Cytometry Assay Kit (Molecular Probe, Life Technologies, Carlsbad, CA, USA), as previously described [38]. Briefly, the cell-permeable CellROX Deep Red reagent is essentially non-fluorescent while it is in a reduced state, but exhibits a strong fluorogenic signal upon oxidation, providing a reliable measure of ROS in live cells [41]. After incubation of the cells with TBHP, the CellROX Deep Red reagent at a final concentration of 1000 nM was added to the cells for 45 min at 37 °C, and then immediately analyzed by flow cytometry. 

### 2.6. Incubation of Wine Extract in Cultured Cells

To explore the effect of wine extract in counteracting oxidative stress, increasing concentrations (from 200 to 800 μg/mL of total polyphenols) of wine extract were added to TPH-1 cells and cardiomyocytes for 6 h, before the addition of TBHP. As a positive control, THP-1 cells and cardiomyocytes were preincubated in some experiments with increasing concentrations (1–10 μM) of the isothiocyanate sulforaphane (Sigma Chemical Co, St. Louis, MI, USA) known to activate the Keap1-Nrf2 signaling pathway [42]. Early apoptosis and cell viability was determined using the Annexin V-FITC Kit (Bender MedSystems GmbH, Vienna, Austria) and 7-amino-actinomycin D (BD Biosciences, Buccinasco, Italy) by flow cytometry. Endotoxin contamination of cell cultures was routinely excluded with the chromogenic Limulus amebocyte lysate assay (Sigma-Aldrich, Milan, Italy).  

### 2.7. RNA Isolation and Quantitative Real-Time PCR

Total RNA was isolated with RNEasy Mini Kit (Qiagen, Hilden, Germany). The concentration and quality of RNA were evaluated using the RNA 6000 Nano LabChip Kit (Agilent 2100 Bioanalyzer, Agilent Technologies Inc., Santa Clara, CA, USA). Reverse transcription of total RNA was carried out using the IScript cDNA Synthesis Kit (Bio-Rad, Hercules, CA, USA), according to the manufacturer’s recommendations. Reverse transcription was performed using the IScript cDNA Synthesis Kit (Bio-Rad, Hercules, CA, USA). The relative mRNA expression levels of Nrf2, HO-1, and GCLC subunit were performed in triplicate using the QuantiTect Primer Assay and QuantiTect SYBR Green PCR Kit (Qiagen) on the MyiQ Thermal Cycler (Bio-Rad), as previously described [38]. QuantiTect Hs-ACTB Assay (Qiagen) was used as the normalizer. Normalized gene expression levels were given as the ratio between the mean value for the target gene and that of the β-actin in each sample. 

### 2.8. Nuclear Assay of Nrf2

Nuclear Nrf2 was measured using sandwich enzyme-linked immunosorbent assay (ELISA) kits (LifeSpan BioScience, Inc., Seattle, WA, USA), following the manufacturer’s instructions. Nuclear extracts were prepared using the Nuclear Extraction Kit (Cayman, Ann Arbor, MI, USA). 

### 2.9. Western Blot Analysis

Western blot analysis was performed as previously described [38]. Briefly, cytoplasmic extracts were prepared using the Nuclear Extraction Kit (Cayman Chemical Company, Ann Arbor, MI, USA), and the protein concentration was determined using the Pierce™ BCA Protein Assay Kit (Thermo Fisher, Waltham, MA, USA). Protein mixtures (70 µg) were fractionated by SDS-PAGE into 10% acrylamide/bis acrylamide gel. Samples were dissolved in Laemli Buffer and boiled for 10 min at 99 °C. Gel was electro-transferred to a nitrocellulose membrane with a Trans-Blot Turbo Blotting system for mini-gel. The membrane was treated with a blocking solution (5% nonfat dry milk in PBS) for 2 h at room temperature on a shaking plate. Then, the membrane was incubated in 5% *w*/*v* nonfat dry milk in PBS, 0,2% tween 20, with anti-GCLC antibody (ab190685, Abcam, UK), diluted 1:1000 overnight at 4 °C; anti-HO-1 antibody (ab52947, Abcam, UK), diluted 1:150 overnight at 4 °C; and anti-beta actin (GTX109639, GeneTex, USA), diluted 1:1000 for 2 h at room temperature. Incubation was followed by three washes and then the membrane was incubated with anti-rabbit secondary antibody (1:5000) for 45 min at room temperature. Semi-quantitative analysis was performed with Quantity One software (BioRad, 1-D Analysis Software) using Adjusted Volume (OD × mm^2^) as the quantitative parameter. The housekeeping protein, Anti beta actin (GTX109639, GenetTex, Irvine, CA, USA), was used to normalize data.

### 2.10. Statistical Analysis

Data were expressed as mean ± SD values if normally distributed. Differences between groups were analyzed by a two-tailed paired and unpaired Student’s t -test and by a one- way analysis of variance followed by the post hoc Tukey test for multiple comparisons. A probability value (*p*) of 0.05 was considered to be statistically significant. All data were analyzed with SPSS (IBM Corp. SPSS Statistic Version 20) (IBM Corp., New York, NY, USA). The raw data derived from UPLC-MS analysis were processed with Progenesys QI (Nonlinear dynamics, UK). The data matrix obtained by using Progenesys QI was submitted to PCA through SIMCA 13.0 (Umetrics, Sartorius, Gottingen, Germany) after Pareto scaling and centering.

## 3. Results

### 3.1. Accumulation of Wine Metabolites in Cultured Cells

In a preliminary experiment, different amounts of cells were treated with wine extract to assess which wine metabolites were bioavailable in THP-1 cells. The extracts from treated and control cells and the wine extract were analyzed by an untargeted metabolomics approach, based on LC-MS. The feature quantification matrix, with a relative amount of each metabolite in each sample, was explored by principal component analysis (PCA) (Figure 1). This analysis clearly separated the wine-treated from the control cells along the first principal component (Figure 1a). The loading plot shows the metabolites responsible for the separation of control and treated cells (Figure 1b). In Appendix A, the list of the metabolites that characterize the treated and control cells, based on p(1) loading values of PCA from Figure 1, are displayed. The relative amount of each metabolite in all the cell samples are shown in a heat map. The wine metabolites found in treated cells belonged mainly to stilbenes and flavan-3-ols derivatives, which represented the more abundant metabolites in the special wine extract used to treat cells. Other metabolites in the treated cells were not typical wine metabolites, and may have been cell catabolites of wine molecules or endogenous cell metabolites induced by the treatment. Wine exposure also caused a decrease in a small group of mostly unidentified metabolites. The accumulation of the wine metabolites within the cells was not directly proportional to the abundance of metabolites within the extract. For instance, the resveratrol tetramer (id. 1) accumulated much more than the procyanidin P3 type (id.17) and procyanidin P2 type (id.29), which were more abundant in the wine. 

The dynamic of metabolite accumulation in THP-1 cells was then followed in a time-course experiment (3 and 6 h). There were 74 different metabolites found in cells treated for 6 **h** with wine extract, which was at least four times more abundant than in the untreated control cells. Of these, 26 were putatively identified and corresponded to typical wine metabolites belonging to the classes of stilbenes, flavan-3-ols and derivatives, and flavonoids; hydroxycinnamic and hydroxybenzoic acids were also detected (Figure 2 and Appendix A).

### 3.2. Effect of Different Degrees of Oxidative Stress on Intracellular ROS Formation and Nrf2 Nuclear Translocation 

We performed dose–response experiments with TBHP to evaluate the effect of different degrees of oxidative stress on ROS formation. Our results showed that TBHP dose–dependently increased ROS generation (*p* < 0.01) in THP-1 cells (Figure 3a) and in cardiomyocytes (data not shown). We then assessed whether different degrees of oxidative stress could affect Nrf2 activation and found that, in the presence of mild oxidative stress, Nrf2 nuclear translocation was induced, whereas the highest concentration of TBHP able to significantly trigger ROS production also significantly blunted Nrf2 nuclear translocation (Figure 3b). 

### 3.3. Polyphenol Content in Red Wine and Effect of Wine Extract and Sulforaphane on TBHP-Induced ROS Generation

As we found a marked accumulation of specific wine metabolites in cultured cells incubated with wine extract, we next assessed whether these intracellular metabolites could counteract ROS generation. First, we measured the concentrations of phenolic compounds present in the red wine utilized in this study. As shown in Table 1, the particular choice of grapes and oenological techniques allowed to achieve a red wine with a particularly high content of polyphenols (3820 ± 123 mg/L). Accordingly, the concentrations of total flavonoids (+ catechin), non-anthocyanin flavonoids (+catechin), (+)catechin, and (−) epicatechin were quite elevated. 

Then, we performed some dose–response experiments with the wine extract (containing from 200 to 800 μg/mL of total polyphenols). Our results indicated that the pre-incubation of cells with increasing concentrations of wine extract determined a dose–dependent reduction in ROS of THP-1 cells and cardiomyocytes (*p* < 0.01) (Figure 4a,b); unsurprisingly, sulforaphane also dose–dependently reduced ROS formation both in TPH-1 cells and cardiomyocytes (*p* < 0.01), (Figure 4a,b). A representative flow cytometer analysis of ROS generation induced by TBHP and the inhibitory effect of the highest dose of wine extract is displayed in Figure 4c. Moreover, our experimental conditions did not affect cell viability, as shown in a representative flow cytometer analysis (Figure 4d).

### 3.4. Effects of Wine Extract on Nrf2/Antioxidant Response Element (ARE) Pathway Activation 

Subsequently, we conducted experiments aimed to assess the effect of wine metabolites on Nrf2/ARE pathway activation. Our results showed that the incubation of cultured cells with wine extract triggered Nrf2 nuclear translocation both in THP-1 cells (Figure 5a) and cardiomyocytes (Figure 5b) (*p* < 0.01). Unsurprisingly, the effect of sulforaphane was even more pronounced (*p* < 0.001), as shown in Figure 5a,b. Thus, we designed further experiments to evaluate whether wine extract could also restore the Nrf2 pathway in cells exposed to oxidative stress. When TBHP was incubated with THP-1 cells and cardiomyocytes, we observed a significant decrease (*p* < 0.01) in nuclear Nrf2 protein when compared with control (Figure 5a,b). On the contrary, pre-incubation with wine extract and sulforaphane was able to counteract the negative effect of TBHP on nuclear Nrf2 (*p* < 0.01), (Figure 5a,b). In addition, the expression (mRNA and protein) of target genes, HO-1 and GCLC, was significantly (*p* < 0.01) upregulated by wine extract in cardiomyocytes (Figure 5c,d), indicating its ability to contrast the unfavorable effect of oxidative stress on antioxidant gene expression (*p* < 0.01).

## 4. Discussion

The results of this study showed that the untargeted metabolomics approach based on UPLC-MS analysis clearly separated the wine-treated cells from control cells. Even though some metabolites in the cells exposed to wine extract were not typical of wine constituents and could be cell catabolites of wine molecules or endogenous cell metabolites induced by the wine extract treatment, metabolites found in treated cells belonged mainly to stilbene and flavan-3-ols derivative groups. Among the 74 different metabolites found in the cells exposed to wine extract in a time course experiment, which was at least four time more abundant than in untreated control cells, only 26 were recognized as typical wine metabolites belonging to the classes of stilbenes, flavan-3-ols and derivatives, and flavonoids; hydroxycinnamic and hydroxybenzoic acids were also identified. Interestingly, the results of this study also showed that the accumulation of wine metabolites in cells exposed to wine extract was not related to the metabolite quantity within the wine extract, and resveratrol, for instance, accumulated much more than the procyanidin P3 type and procyanidin P2 type in spite of their higher abundance in our wine extract. Thus, the rate of accumulation and/or the metabolite catabolysis within the cells were metabolite-specific. From a chemical point of view, polyphenols, in general, and those present in red wines, in particular, are compounds possessing a phenyl ring carrying one or more hydroxyl groups, which can reduce ROS and confer strong antioxidant activity [43]. Even though these compounds are scarcely bioavailable in vivo and it is unlikely that their health effects are solely the result of direct antioxidant activity, clinical studies have also reported that the beneficial effects of polyphenols may be dependent on the amount of phenolic compounds ingested [44]. In general, the absorbable monomeric polyphenols and oligomeric tannins in in vivo situations are subjected to three main types of conjugation: methylation, sulfation, and glucuronidation [13]. Even if conjugation seems to reduce the specific activities of polyphenols [45], the possibility exists that these polyphenols are deconjugated at the cellular level, and consequently work at the tissue level as the original molecules [46]. In this context, polyphenols have been detected in a wide range of tissues in mice and rats, including the brain, endothelial cells, heart, kidney, spleen, pancreas, prostate, uterus, ovary, mammary gland, testes, bladder, bone, and skin [13]. On the basis of these considerations, it is likely that the accumulation of specific wine metabolites in cells resembles what occurs in animal and human tissues after polyphenol absorption. Although the experimental approach used in this study cannot be representative of an in vivo situation, the results of our research show that there was a marked and selective accumulation of specific wine metabolites in cultured cells incubated with wine extract. Even though the significance of this accumulation has to be studied in depth, the fact that at least a part of the metabolites present in our wine extract and in cells belong to the class of absorbable monomeric polyphenols and oligomeric tannins support the concept that some wine metabolites could directly interfere with cellular signaling in vivo. Interestingly, our results also showed that the accumulation of specific wine metabolites in cells was associated with a dose–dependent reduction in TBHP-induced ROS generation in cells probably related, but not only, to the radical scavenger activity of the wine metabolites accumulated in the cells. This is a peculiar discovery, as the reduction of oxidative stress was not the result of a single phytochemical activity, but of all the components of wine extract that crossed cell membranes and were present within the cells. Even though we are aware of the approximations that occur when comparing in vivo and in vitro conditions, the fact that the concentration of polyphenols used in our experiments was similar to that contained in a glass (150 mL) of our red wine reinforces the importance of these results. Besides quenching free radicals through their radical scavenging potential, some polyphenols have a positive effect on antioxidant gene expression, preventing chain reactions and the activation of oxygen to highly reactive products [24]. One of the crucial cellular defense mechanisms triggered by polyphenols against oxidative stress is the Nrf2/ARE pathway [24,26]. Under a normal state, Nrf2 is kept dormant in the cytoplasm by binding to Keap1. Under different conditions, the Nrf2/Keap1 complex can dissociate, leading to Nrf2 entry into the nucleus, where it binds to ARE, activating a series of genes correlated with the antioxidant response [26]. In this context, the data of this study show that the wine extract modified Nrf2 pathway activity, giving rise to an elevation of nuclear Nrf2 translocation in THP-1 and cardiomyocyte cells. This indicates that the polyphenols contained in the wine extract that crossed the cell membranes and were present within the cells were able to dissociate the Nrf2/Keap1 complex, favoring Nrf2 translocation into the nucleus. In this context, however, the modulation mechanism of Keap1-Nrf2/ARE signaling pathway activation can be roughly divided into two categories: Keap1-dependent and Keap1-independent mechanisms [47]. The modification (e.g., chemical adduction, oxidation, nitrosylation, or glutathionylation) of one or more critical cysteine residues in Keap1 is likely a chemical–biological trigger for the activation of Nrf2 [47]. However, an increasing body of literature has revealed alternative mechanisms of Nrf2 regulation, including phosphorylation of Nrf2 by various protein kinases, interaction with other protein partners, and epigenetic factors [47]. As extensively reviewed by Zhou Y et al. [48], it is likely that most polyphenols activate the Nrf2/ARE pathway in a Keap1-dependent manner, even though a small group of these compounds can also operate in a Keap1-independent manner, or both. 

We used the highest concentration of TBHP to trigger ROS generation within the cells, as this concentration led to a significant decrease in nuclear Nrf2 protein translocation into the nucleus, both in THP-1 and cardiomyocyte cells. Although this finding may appear paradoxical, the Nrf2 response to oxidative stress has been reported to be biphasic and repressed when oxidative stress is relatively high [32]. Oxidative stress, in fact, has been reported to activate I-kB kinase, which phosphorylates NF-kB, leading to its translocation into the nucleus and activation of pro-inflammatory cytokines [32]. Furthermore, when NF-kB binds to cAMP-response-element binding proteins (CBP) in a competitive manner, it inhibits the binding of CBP to Nrf2, which leads to the inhibition of Nrf2 transactivation [32]. In addition, NF-kB has been shown to increase the recruitment of histone deacetylase to the ARE region; hence, Nrf2 transcriptional activation is prevented [32]. In a recent study published by our group, the oxidative stress induced by ischemia-reperfusion caused a significant increase in nuclear phosphorylated p65, an indicator of NF-kB activation, which supports the current findings [38]. Characterized by relatively high oxidative stress with consequent inhibition of Nrf2 transactivation, the wine extract counteracted the TBHP-induced reduction of Nrf2 translocation and significantly increased Nrf2 entry into the nucleus. Accordingly, the mRNA and protein expression of HO-1 and GCLC in cardiomyocytes were significantly upregulated by wine extract, confirming that wine extract triggers the Nrf2/ARE pathway. Our experimental model may therefore be applicable for studying the effects of specific wine metabolites and of polyphenols on the modulation of cell signaling pathways and their molecular mechanisms. This may lead to a better understanding of the large-scale health protective effects of some of the most important dietary polyphenols. A limitation of our in vitro study was that the clinical benefits of dietary polyphenols also depends on the gut microbiota, which is independently associated with health benefits [49]. In the colon, gut bacteria metabolize non-absorbable polyphenols into bioactive compounds that produce clinical benefits [49]. Our incubation of wine extract with cells did not consider the role of these bioactive compounds. A further limitation was that we did not evaluate Nrf2 binding to target promoters; in particular, a Chip analysis could have confirmed our conclusions. We believe that more studies are needed to assess the effects of main intracellular wine metabolites on cellular signaling and on promoter binding, not only of Nrf2, but also of other transcription factors. 

## 5. Conclusions

In these cellular models we demonstrate that, after incubation with wine extract, particular polyphenol metabolites can accumulate within the cells in a metabolite-specific manner. This accumulation allows the cells to oppose oxidative stress and upregulate the antioxidant Nrf2/ARE pathway. Although these results could help explain the healthy activity of wine polyphenols within the cells, further studies are needed to confirm the potential of monomeric polyphenols and oligomeric tannins in wine extract to limit oxidative stress in vivo.

## Figures and Tables

**Figure 1 antioxidants-11-02055-f001:**
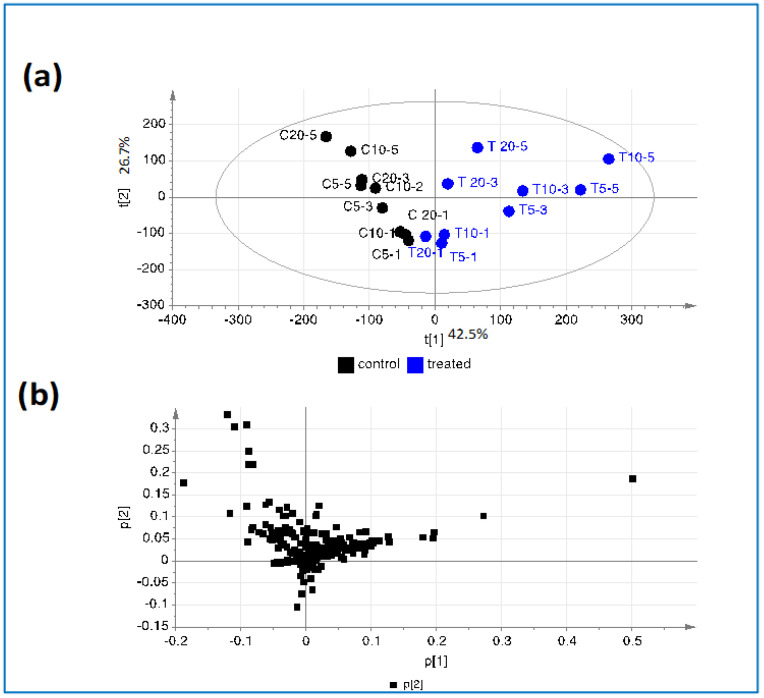
Principal component analysis of control cells (C) and cells pre-incubated with wine extract (T), according to their metabolite levels: (**a**) score plot and (**b**) loading plot. To set up the experimental procedure, 5, 10, and 20 millions of cells were used, and 1, 3, and 5 μL of cell extract were analyzed.

**Figure 2 antioxidants-11-02055-f002:**
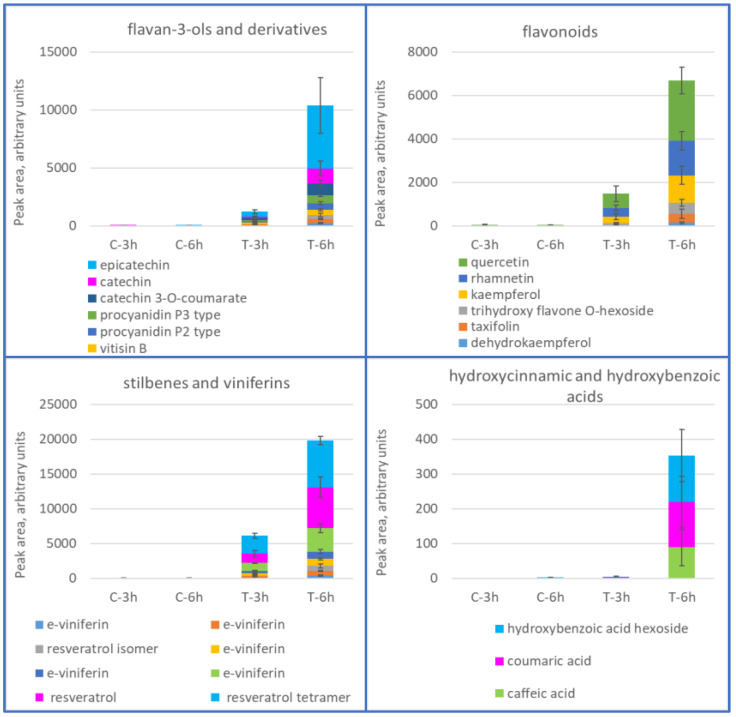
Flavan-3-ols and derivatives, flavonoids, stilbenes/viniferines, and hydroxycinnamic and hydroxybenzoic acids in cells treated with wine extract (T) for 3 and 6 h, and in control cells (C). Mean values of two biological replicates, +/− standard deviation, as determined by UPLC-ESI-MS and expressed in arbitrary units.

**Figure 3 antioxidants-11-02055-f003:**
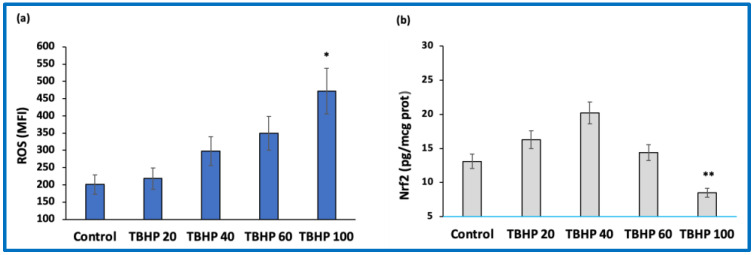
Effect of different degrees of oxidative stress on intracellular reactive oxygen species (ROS) formation and Nrf2 nuclear translocation in THP-1 cells. (**a**) ROS formation in cells exposed to increasing concentrations (from 20 to 100 μM) of tert-butyl hydroperoxide (TBHP). (**b**) Nrf2 nuclear translocation in cells exposed to increasing concentrations (from 20 to 100 μM) of tert-butyl hydroperoxide (TBHP). Data represent the mean +/− standard deviation of measurements performed in triplicate in three different experiments; * *p* < 0.01 increase vs. control; ** *p* < 0.01 decrease vs. control.

**Figure 4 antioxidants-11-02055-f004:**
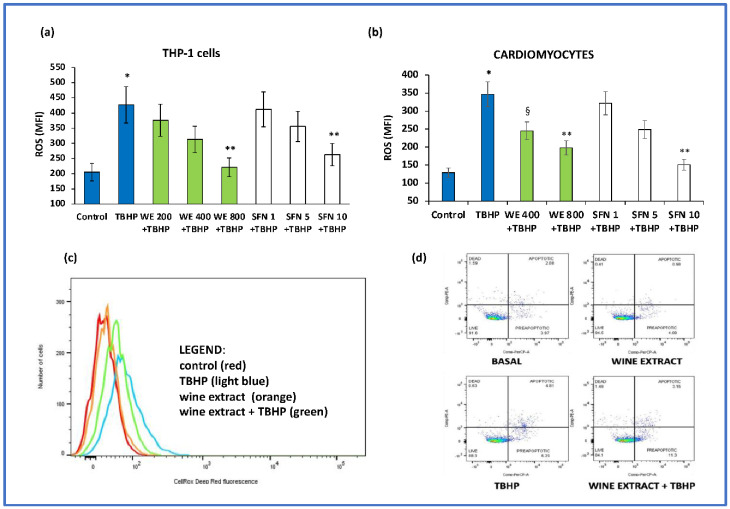
Dose–response effect of wine extract (WE) and sulforaphane (SFN) on oxidative stress-induced intracellular reactive oxygen species (ROS) formation. (**a**,**b**) THP-1 cells and cardiomyocytes were pre-incubated with increasing concentrations (from 200 to 800 μg/mL of total polyphenols) of WE and SFN (1–10 μM) before the addition of tert-butyl hydroperoxide (TBHP, 100 μM). Data represent the mean +/− standard deviation of measurements performed in triplicate in three different experiments; * *p* < 0.01 increase vs. control; ** *p* < 0.001 decrease vs. THBP; ^§^
*p* < 0.1 decrease vs. TBHP. (**c**) Representative flow cytometry analysis of ROS generation induced by TBHP and the inhibitory effect of the highest dose of wine extract. (**d**) Representative flow cytometry analysis on cell viability of the highest dose of wine extract.

**Figure 5 antioxidants-11-02055-f005:**
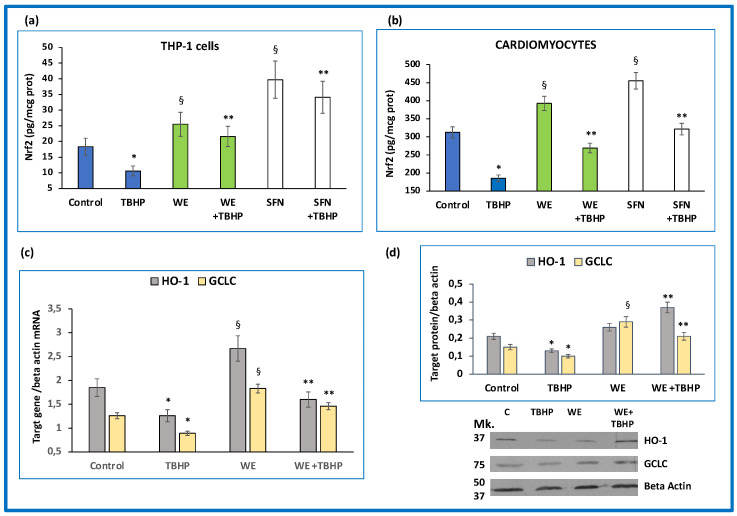
Wine metabolites activate Nrf2/ARE pathway in basal conditions and in cells exposed to oxidative stress. (**a**,**b**) Effect of wine extract (WE) and sulforaphane (SFN) on Nrf2 nuclear translocation. THP-1 cells and cardiomyocytes were pre-incubated with WE, containing 800 µg/mL of total polyphenols and 10 μM SFN with or without the addition of tert-butyl hydroperoxide (TBHP, 100 µM). (**c**) Effect of WE on mRNA expression of Nrf2 target genes hemeoxygenase-1 (HO-1) and glutamate-cysteine ligase catalytic subunit (GCLC). (**d**) Average quantification of HO-1 and GCLC protein expression obtained by densitometric analysis and representative Western blot analyses for HO-1 and GCLC protein expression in cardiomyocytes. Data represent the mean +/− standard deviation of measurements performed in triplicate in three different experiments and; * *p* < 0.01 decrease vs. control; ^§^
*p* < 0.01 increase vs. control; ** *p* < 0.01 vs. TBHP.

**Table 1 antioxidants-11-02055-t001:** Concentrations of the main phenolic compounds in red wine “*Vitis Vitae*”.

Phenolic Compounds	
Total polyphenols (mg/L)	3820 ± 117
Total flavonoids (+catechin) (mg/L)	2452 ± 92
Non-anthocyanin flavonoids (+catechin) (mg/L)	2153 ± 97
(+) Catechin (mg/kg)	62.5 ± 5.3
(-) Epicatechin (mg/kg)	49.3 ± 4.1

## Data Availability

The data presented in this study are available on request from the corresponding Author.

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
