# Peer review of "Intracellular Polyphenol Wine Metabolites Oppose Oxidative Stress and Upregulate Nrf2/ARE Pathway"

_antioxidants, 2022, doi:10.3390/antiox11102055_

Round 1

Reviewer 1 Report

Dear authors,

After the review process, I have several comments: you should include more numerical data in the abstract; In each Materials and Methods sections, you should add references; starting from the bioactive potential of functional products and bioavailability of phenolic compounds, you should present future valorization of their research; the relation with microbiota is essential to increase the bioavailability of phenolic compounds from wine and the resistance to oxidative stress; the limitation of the study is a secondary part necessary in the discussion section to complete the study.

Best regards!

Reviewer 2 Report

Stranieri et al. have studied the effect of polyphenols extract from a specific wine on oxidative stress and the pathways involved. This topic is far to be original as this point has been investigated for years. Despite that comment, the originality if more in the fact that a specific pathway, controlled by Nrf2, has been analyzed.  

The work is actually divided in two main parts:

* the first is focused on the polyphenolic contents that have been extensively died and documented using clear cut methods. Even interesting, this part is rather classical as it describes the chemical content of the wine extract. 

* the second part has aimed to investigate the Nfr2 pathway. Interestingly the authors have measured 2 Nrf2 "target genes", HO-1 and GCLC, at both RNA and protein levels. Results are clear enough to support the conclusions.

To increase the quality of the molecular study, it could be of interest to investigate whether bindings of Nrf2 to the target promoters is increased when the cells are treated with WE and SFN (as a positive control). This would definitively confirm the author's conclusion. This could be easily performed with a ChiP analysis.  

Round 2

Reviewer 1 Report

No other comments.

Author Response

Dear Reviewer,

thank you for accepting our revision.

Reviewer 2 Report

Authors have adequately answered the various comments. Regarding the effects of polyphenols on binding of NRF2 to various target promoters, this should at least be discussed by the authors as a putative lack of the study and/or a potential experiment to perform in the future.

Author Response

Dear Reviewer,                                                                                                              we have considered your comment concerning the effects of polyphenols on binding of Nrf2 to various promoters. As you suggested, we have now added in the Discussion that is a limitation of our study and an important experiment to perform in future researches (page 12, from line 470 to line 474).